# Atherosclerosis Burdens in Diabetes Mellitus: Assessment by PET Imaging

**DOI:** 10.3390/ijms231810268

**Published:** 2022-09-06

**Authors:** Poul F. Høilund-Carlsen, Reza Piri, Per Lav Madsen, Mona-Elisabeth Revheim, Thomas J. Werner, Abass Alavi, Oke Gerke, Michael Sturek

**Affiliations:** 1Department of Nuclear Medicine, Odense University Hospital, 5000 Odense, Denmark; 2Research Unit of Clinical Physiology and Nuclear Medicine, Department of Clinical Research, University of Southern Denmark, 5230 Odense, Denmark; 3Department of Cardiology, Herlev Gentofte Hospital, 2900 Herlev, Denmark; 4Division of Radiology and Nuclear Medicine, Oslo University Hospital, 0424 Oslo, Norway; 5Institute of Clinical Medicine, Faculty of Medicine, University of Oslo, 0315 Oslo, Norway; 6Department of Radiology, Perelman School of Medicine, University of Pennsylvania, Philadelphia, PA 19104, USA; 7Department of Anatomy, Cell Biology, Physiology, Indiana University School of Medicine, Indianapolis, IN 46202, USA

**Keywords:** atherosclerosis, PET, ^18^F-sodium fluoride (NaF), ^18^F-fluorodeoxyglucose (FDG), inflammation, calcification, quantification

## Abstract

Arteriosclerosis and its sequelae are the most common cause of death in diabetic patients and one of the reasons why diabetes has entered the top 10 causes of death worldwide, fatalities having doubled since 2000. The literature in the field claims almost unanimously that arteriosclerosis is more frequent or develops more rapidly in diabetic than non-diabetic subjects, and that the disease is caused by arterial inflammation, the control of which should therefore be the goal of therapeutic efforts. These views are mostly based on indirect methodologies, including studies of artery wall thickness or stiffness, or on conventional CT-based imaging used to demonstrate tissue changes occurring late in the disease process. In contrast, imaging with positron emission tomography and computed tomography (PET/CT) applying the tracers ^18^F-fluorodeoxyglucose (FDG) or ^18^F-sodium fluoride (NaF) mirrors arterial wall inflammation and microcalcification, respectively, early in the course of the disease, potentially enabling in vivo insight into molecular processes. The present review provides an overview of the literature from the more than 20 and 10 years, respectively, that these two tracers have been used for the study of atherosclerosis, with emphasis on what new information they have provided in relation to diabetes and which questions remain insufficiently elucidated.

## 1. Introduction

Atherosclerosis is a significant co-morbidity in diabetes mellitus, and its sequalae, acute myocardial infarction, stroke, and peripheral artery disease, are major reasons why diabetes has recently entered the top ten causes of death worldwide [1]. However, from the literature, it is not easy to determine the actual incidence of atherosclerosis in diabetes mellitus (DM) and how, how early, or how fast the disease develops in diabetic patients, or to what extent it can be counteracted by drugs or other therapies. One reason for this is that in vivo diagnosis and assessment of atherosclerosis has traditionally been based on indirect implications, such as increased pulse wave velocity (PWV) in large arteries or arterial (typically carotid) intima-media thickness or on computed tomography (CT). Unfortunately, CT depicts late occurring arterial wall macrocalcification, which must attain a certain size to become detectable [2,3].

This review focuses on the use of molecular imaging with positron emission tomography (PET) in the form of PET/CT to assess atherosclerosis in diabetes. Even though a multitude of PET tracers have been applied, only two have been studied sufficiently to be of interest in this context, i.e., ^18^F-fluorodeoxyglucose (FDG) and ^18^F-sodium fluoride (NaF), which enable the detection and measurement of inflammation and active microcalcification, respectively, in early-stage atherosclerosis, long before CT-detectable calcification is formed [4]. Multiple publications have described the connection between diabetes and atherosclerosis [5,6,7] and the consequences of atherosclerosis in diabetic patients, including changes in cerebral [8,9,10] and cardiac perfusion and function [11,12,13], renal insufficiency [14,15,16], diabetic foot, etc. [17,18,19], topics that all deserve their own review.

We take a molecular perspective in focusing on PET/CT, because the superior sensitivity of PET enables detection of molecular changes long before tissue changes become apparent using CT. The strength of CT is its much higher spatial resolution, which can show more precisely where in the body the molecular changes are taking place [20,21]. In doing this, we have emphasized facets of atherosclerosis in relation to diabetes, i.e., mechanisms, detection, and prevalence, associations with risk factors, progression, and the effects of therapy as assessed by PET/CT imaging.

## 2. Results

Our systematic literature search (see Section 4) yielded 283 hits. After the removal of duplicates and records outside the scope of this review, 65 potentially eligible full-text articles remained, of which 39 were excluded for various reasons (Figure 1). Two studies, discovered from quotes in other included articles, were added, leaving 28 papers for analysis.

Below (and in Appendix A) the literature is summarized according to five categories:Inflammation mechanisms and targeting (FDG only, see below) [22,23,24,25,26,27,28,29],Early detection and prevalence of arterial FDG or NaF uptake [30,31,32,33,34,35],Cardiovascular risk in DM [36,37,38],Disease progression [39,40],Therapy [41,42,43,44,45,46,47,48,49].

Seven studies dealt with the carotids [22,24,25,28,31,33,37], five with the coronary arteries [26,35,40,42,45], three with the aorta [23,29,44], and three with the femoral arteries [32,38,43]. Three studies focused on the carotids and the aorta [41,46,49], two on the coronary arteries and the thoracic aorta [47,48], and five on several major arteries [27,30,34,36,39]. Four were animal experiments [23,26,29,44]. One tracer only was applied in most studies (FDG in 19, NaF in 7), one study used FDG and NaF [39], and one study used FDG and 18F-flouromethylcholine [29]. Unless otherwise stated, mean or median differences presented below are statistically significant.

### 2.1. Inflammation Mechanisms and Targeting of FDG

Three studies were animal experiments from the PET Centre in Turku, Finland. Silvola et al. investigated the effects of age, duration of a high-fat diet, and type 2 diabetes mellitus (T2DM) on atherosclerotic plaque development and the uptake of FDG. Atherosclerotic low-density lipoprotein receptor deficient mice were compared to atherosclerotic and T2DM mice that overexpressed insulin-like growth factor II, and C57BL/6N mice on normal chow. From ages four to six months and twelve months and older, plaque size increased, and the macrophage density decreased. Compared with controls, PET showed increased aortic FDG uptake at four and six months, but not at twelve months and older. Autoradiography demonstrated focal FDG uptake in plaques at all time points with highest uptake in plaques with high macrophage density. There were no differences in plaque size, macrophage density, or uptake of FDG between the two types of mice at any time point [23]. Hellberg et al. studied kinetics and the uptake distribution of ^18^F-flouromethylcholine and FDG in many organs in the same mice models with similar methods, and found that whole aortic FDG uptake was higher than aortic ^18^F-flouromethylcholine uptake in both diabetic (4.7 vs. 2.9) and nondiabetic mice (5.0 vs. 1.5) [29]. Finally, in ten farm pigs with streptozotocin-induced diabetes examined after six months of a high-fat diet, Tarkia et al. found that an increased FDG uptake was present by ex vivo counting and autoradiography in coronary atherosclerotic lesions, although not detectable by prior in vivo PET in these early stage lesions [26].

Among the six human studies, Kim et al. compared carotid FDG uptake and intima-media thickness in three age- and sex-matched samples, each of 30 subjects: one with T2DM, another with impaired glucose tolerance, and a third with normal glucose tolerance. This study also examined associations with the Framingham Risk Score, high sensitivity C-reactive protein (hsCRP), adiponectin, and anti-inflammatory adipokine. They measured the maximal standardized uptake value (SUVmax) and the target-to-background ratio (TBRmax), i.e., the SUVmax divided by blood pool radioactivity, and found that both parameters were slightly higher in the sample with impaired glucose tolerance, and clearly higher in T2DM patients than in normal glucose tolerance subjects. Meanwhile, there were no significant differences with regard to carotid intima-media thickness. TBRmax values had the strongest positive correlation with hsCRP levels among cardiovascular risk factors, but other inflammatory markers were not associated with TBRmax values. Their conclusion was that vascular inflammation measured by FDG PET was increased in healthy individuals without hyperlipidemia, but with elevated hsCRP, even if the increases of SUVmean and TBRmean (+11% and +18%, respectively) in the impaired glucose tolerance group were very modest compared to the increases in the T2DM group (+43% and +60%, respectively) [22]. Yang et al., from the same Korean institution, studied FDG uptake in the right carotid arteries of 41 patients with T2DM of unknown duration and of 41 healthy individuals, and its association with soluble forms of the receptor for advanced glycation end-products (RAGE), including the splice-variant endogenous secretory RAGE. The TBRmean and TBRmax values were higher (+62% and +94%, respectively) in T2DM patients compared to healthy subjects. Circulating soluble RAGE and endogenous secretory RAGE concentrations were insignificantly lower in the T2DM group. After adjusting for age and gender, soluble RAGE levels were significantly negatively correlated (large scatter) with both mean and maximum TBR values, but not with carotid intima-media thickness values [24].

Bucerius et al. [25] examined the impact of non-insulin dependent T2DM on carotid wall FDG uptake in 43 patients with non-insulin dependent T2DM vs. 91 nondiabetic subjects, of whom 86% and 74%, respectively, were on statin medication. All had documented or suspected cardiovascular disease (CVD). SUVmax and TBRmax in the common carotid artery were given after “correction” for higher blood glucose levels in the diabetic patients (mean 6.8 mmol/L vs. 5.4 in nondiabetic subjects) using a formula originally suggested by the European Association of Nuclear Medicine (EANM) in a position paper on oncologic FDG PET imaging [50]. The glucose-corrected values were all about 20% (and significantly) higher in the T2DM group, whereas the uncorrected values were all marginally (and insignificantly) lower. Based on “corrected” and hence increased values, the authors concluded that T2DM is significantly associated with carotid wall FDG uptake in patients with known or suspected CVD disease. Moreover, they found that, in diabetic patients, obesity and smoking add to the risk of increased FDG uptake [25].

The potential relationship between FDG uptake in major arteries and arterial stiffness assessed as central systolic blood pressure (cSBP), carotid-femoral PWV, and the augmentation index was examined by De Boer et al. in 44 patients with early T2DM (mean duration one year) without CVD and anti-diabetic medication. Coronary arteries were excluded in the study because of the large FDG uptake due to cardiac metabolism. SUVmax was normalized for blood glucose levels, as described above, and CT-detectable arterial calcification scored from 0 to 4 (no calcification to calcified plaque involving >50% of vessel circumference) were endpoints. Corrected TBRmax was significantly associated with PWV (R = 0.47, *p* = 0.001) and cSBP (R = 0.45, *p* = 0.003), but not with the augmentation index. The TBRmax of each separate segment was also significantly associated with PWV and cSBP. In a multiple linear regression model including age, sex, BMI, hemoglobin A1c (HbA1c), hsCRP, cholesterol, cSBP, and PWV, PWV was the strongest determinant of mean TBRmax. Seven patients had a score of 0, and seven scored 4 for calcification. The latter was only present in the abdominal aorta and iliac artery. TBRmax was lowest in the carotids, highest in the caudal parts of the aorta, and intermediate in the iliac and femoral arteries [27].

Honda et al. published the only study on the relationship between flow-mediated dilation of the brachial artery, assessed by ultrasound, and carotid vascular inflammation, assessed by FDG PET/CT. Of 145 examined patients, only 19 had T2DM. Multiple logistic regression analysis revealed that age, male sex, low-density lipoprotein-cholesterol and carotid TBR values, but not diabetes, were independently associated with %dilation. In 33 drug-naïve patients with essential hypertension who were included in the study, the changes from the baseline of %dilation after antihypertensive treatment for six months were correlated with those of carotid TBRmax. There was an inverse correlation between Δ%dilation and ΔTBRmax, and ΔTBRmax was the sole independent associate of Δ%dilation in hypertensive patients. In the 19 T2DM patients, mean TBRmax and mean HbA1c values were within normal ranges, but HbA1c values correlated with decreased %dilation [28].

### 2.2. Early Detection and Prevalence of Arterial FDG or NaF Uptake

It seems that Yun et al. were the first to report arterial FDG uptake in diabetic patients. Using visual inspection, they looked for FGD uptake in the abdominal aorta and iliac and proximal femoral arteries of a heterogeneous group of 156 patients, many of whom had various atherogenic risk factors referred to PET for various reasons. Of only 13 patients with diabetes (of unspecified type and duration), 9 (69%) had aortic and iliac artery uptake and 12 (92%) had femoral artery FDG uptake. Patients with ≥1 cardiovascular risk factor (*n* = 133) had, in general, higher visual uptake in all three arterial beds than patients without risk factors (*n* = 23): femoral 70% vs. 22%, iliac 54% vs. 30%, abdominal aorta 53% vs. 35% [30].

A study by Tahara et al. focused on the relation between serum advanced glycation end products (AGEs) and vascular inflammation in 275 outpatients with mean fasting blood glucose of 103.2–107.2 mg/dL and HbA1c 5.91–6.0%. They further focused on 18 patients whose AGE value was >14.2 units/mL, examining whether changes in AGE levels after treatment with oral hypoglycemia agents (OHAs) correlate with those of TBR. It did not provide much information about FDG uptake in diabetes, although carotid FDG uptake was measured in prediabetic and a few treated T2DM patients. The serum AGE level was independently associated with carotid TBRmax and pioglitazone, but not with glimepiride, reduced AGE, or HbA1c levels. TBRmax values were not compared to those of a relevant control group; nor were they given for the 18 treated patients. It was only stated that the ΔAGEs obtained by OHA treatment correlated positively with ΔTBR [31].

Bernelot Moens et al. measured carotid FDG uptake in patients with peripheral artery disease (PAD) defined by their ankle-brachial index (<0.9 and/or decreased >0.15 after a treadmill test). The study compared 11 controls with 11 patients with PAD but no diabetes, 11 with PAD+T2DM, all on oral therapy (T2DM), and 12 with PAD+T2DM on combined oral/insulin (insulin-dependent) therapy; it was found that blood-glucose-normalized TBRmax was 32% higher in the first group, 95% higher in the second, and 122% higher in the third group, despite comparable PAD severity, BMI, and CRP in the three groups. Multivariate regression analysis showed that HbA1c and plasma insulin levels correlated with TBR, but the dose of exogenous insulin did not [33].

Bural et al. studied 110 oncologic patients imaged with FDG PET/CT, 55 insulin dependent diabetes patients, and 56 comparable nondiabetic controls; they observed that the average blood-glucose-corrected SUVmax and SUVmean valules were significantly higher (27% and 20%, respectively) in insulin-dependent subjects in all arterial segments (aorta, iliac, and femoral arteries). Calcification by CT was 27% higher, and both FDG uptake and calcification were most pronounced in the arch and abdominal part of the aorta and in the iliac arteries [34].

Finally, only two studies dealt with arterial NaF uptake. In an early report, Janssen et al. noted by visual inspection the “linear” NaF uptake in the femoral arteries of 159 (39%) of 409 oncologic patients, 44 of whom had unspecified diabetes. Among these, 43 had calcification by CT, of which 28 were NaF positive, while only one was negative according to both CT and NaF PET [32]. In contrast to several aforementioned FDG findings, Raggi et al. studied 88 consecutive, well-controlled ambulatory patients with long lasting (median 15 years) diabetes (48 T2DM + 40 insulin-dependent), who were all asymptomatic for CVD and had a mean HgbA1c of 7.9%. There was increased coronary artery NaF uptake in only 13 patients (15%), 4 of whom did not have co-localized CT-detectable calcification, suggesting slow atherosclerosis progression in well-controlled diabetes patients [35].

### 2.3. Cardiovascular Risk in Diabetes

Strobl et al. analyzed 315 consecutive patients scanned with PET/CT for non-cardiovascular indications, such as malignant tumors, to study the impact of age, gender, and cardiovascular risk factors on vessel wall inflammation (FDG uptake) and the calcified plaque burden (by CT) in the common carotids, thoracic and abdominal aorta, and iliac arteries. Only 15 patients had diabetes (fasting glucose ≥ 200 mg/dL or treatment with a hypoglycemic agent). The calcified plaque score was associated with diabetes in all four vascular beds and, overall, was more closely associated with cardiovascular risk factors compared to TBR, which was not associated with diabetes in any of the four vascular beds. In the aorta, TBRmax was associated with age ≥ 65 years and male gender; in the carotids, it was also associated with a BMI ≥ 30, but in the iliac arteries with age only [36].

Lee et al. [37] analyzed the scans of 290 asymptomatic adults tested with FDG PET/CT scans as part of a health screen, of whom 92 had metabolic syndrome (MetS), according to American Heart Association criteria. They examined the relation between carotid artery FDG uptake and the Framingham Risk Score and evaluated the possible role of FDG uptake in terms of the risk stratification of asymptomatic adults. Pre-scan blood-glucose-corrected TBRmax values, termed “TBRglu”, were called “high” if ≥1.5 corresponded to the 75th percentile in all 290 subjects. TBRglu was 15% higher in MetS than non-MetS patients (1.5 vs. 1.3, *p* < 0.001), contrary to the findings of Strobl et al. [36]. Carotid FDG uptake was significantly associated with risk factors. Triglyceride levels, diabetes, and MetS were independent determinants of high TBRglu. MetS subjects with high carotid FDG uptake had significantly higher levels of hsCRP. Framingham Risk Scores were significantly higher in those exhibiting high, instead of low, carotid FDG uptake, in both subjects with MetS (21.8 ± 16.0 vs. 13.5 ± 11.9) and without MetS (13.1 ± 7.0 vs. 8.2 ± 7.4) [37].

Takx et al. studied 68 patients with T2DM and arterial disease, defined as an ankle-brachial index of <0.9 as part of a randomized controlled trial on the effect of vitamin K2 on T2DM. They looked for potential determinants of modifiable determinants of NaF uptake in the femoral arteries, and found that higher CT calcium mass, total cholesterol, and HbA1c were associated with higher NaF TBRmax [38].

### 2.4. Disease Progression

Reijrink et al. analyzed arterial FDG and NaF uptake in 10 patients with early T2DM from the so-called RELEASE trial (on the effect of linagliptin on arterial FDG uptake) to investigate—as the authors put it—the “prospective correlation” between tracers over time (five years) and whether they are “prospectively (FDG) and retrospectively (NaF) related to progression of systemic arterial disease”. Pre-scan glucose-normalized FDG and NaF TBRmax values were measured in the carotids, three parts of the thoracic aorta, and in the iliac and femoral arteries using different generation PET scanners on two separate occasions. After five years, the arterial calcified plaque score, which was assessed visually based on low dose CT scans (CPscore), had increased by 24%, whereas the measured PWV remained stable (+7.2%), as did many other parameters. The baseline mean FDG TBRmax demonstrated a strong positive correlation with five-year mean NaF TBRmax, as did the latter with the five-year CPscore. Baseline FDG uptake did not correlate with age, the baseline CPscore, the follow-up CPscore, the follow-up PWV, or the ΔCPscore. The CPscores at baseline and at follow-up were strongly correlated. Follow-up NaF uptake correlated positively with the baseline and five-year CPscores and ΔCPscores, whereas it was not associated with %change in CPscore, PWV, or ΔPWV. The authors concluded that “systemic arterial inflammation is an important pathogenetic factor in systemic arterial microcalcification development” [39]. The data certainly show the predictive value of FDG TBRmax for microcalcification, but evidence for causation will require an intervention that blocks FDG uptake.

Bellinge et al. examined 41 patients with diabetes of a duration of 8–11 years and no history of coronary artery disease; they focused on whether localized coronary artery NaF uptake “predicts” development of new CT-detectable calcifications at least two years later. The patients were T2DM, and about 2/3 received statins, and nearly 1/3 insulin; all had previously undergone coronary calcium score (CCS) screening using CT and NaF PET/CT as baseline assessments for a trial on the arterial effects of vitamin K1 and colchicine in subjects with diabetes (see also [47]). Rescans two years later showed that the proportion of “CCS progressors” was higher among NaF-positive than NaF-negative coronaries at baseline (86.5% vs. 52.3%, *p* < 0.001), that NaF-positive disease was an independent “predictor” of subsequent CCS progression (odds ratio 2.92 [95% CI 1.32–6.45], *p* = 0.008), and that all subjects (15/15) with ≥2 NaF-positive coronary arteries progressed in CCS [40]. 

### 2.5. Therapy

Three out of six FDG studies in this section came from the same institution in Japan and presented therapeutic data. Two of them came from the same randomized clinical trial examining the effect of four-month treatment of patients with impaired glucose tolerance or T2DM, all of whom had ultrasound evidence of carotid atherosclerosis and FDG uptake in carotid plaque. In a study by Mizuguchi et al., 31 patients were assigned to pioglitazone (15–30 mg/d) and 21 to glimepiride (0.5–4 mg/d), and completed the trial with two PET/CT scans. Plaque TBRmax, uncorrected for blood glucose levels, was determined as the average of the common carotid arteries and the ascending aorta. The treatments reduced fasting plasma glucose and HbA1c values comparably, but only pioglitazone decreased atherosclerotic plaque inflammation, from 1.46 to 1.32 (*p* < 0.01), vs. an insignificant increase with glimepiride, from 1.35 to 1.41 (Figure 2). Compared with glimepiride, pioglitazone significantly increased high-density lipoprotein cholesterol levels, an increase which was independently associated with the attenuation of plaque inflammation. It also decreased hsCRP, whereas glimepiride did not [41].

Nitta et al. studied patients from the same trial, leaving out those with a visually graded myocardial FDG uptake ≥2 out of a maximal score of 3. Twenty-five pioglitazone-treated patients were compared to 22 glimepiride-treated patients. After a 16-week treatment course, fasting plasma glucose and glycosylated hemoglobin values were comparably reduced in both groups. The reduction in TBRmax from the baseline was significant for the pioglitazone group (from 1.3 to 1.26; *p* = 0.033), whereas there was an insignificant increase in that of the glimepiride group (from 1.45 to 1.54; NS). Similar changes were observed with regard to hsCRP. Neither drug affected the vascular remodeling of the coronary arteries, as assessed by vessel diameter or calcification score [42]. The presence of a significant decrease in FDG uptake, but no decrease in vascular remodeling, leads to the inference that either FDG uptake does not measure inflammation, or that inflammation does not have a causal role in atherosclerosis.

Finally, in a sub-study of the Honda trial [28], Tahara et al. focused on 38 T2DM patients with carotid atherosclerosis who had already received OHAs, except for pioglitazone. All underwent a 75 g oral glucose tolerance test, blood chemistry analysis, and FDG-PET/CT. FDG uptake was measured in the left main coronary artery and was, in findings opposite to those of Honda et al., expressed as blood glucose corrected. They observed that fasting plasma glucose, 30-, 60-, 90-, 120-min post-load plasma glucose, HbA1c, and the left main trunk of coronary artery (LMT)-TBRmax values were significantly decreased by add-on pioglitazone therapy, whereas high-density lipoprotein-cholesterol and adiponectin levels were increased. Increased serum levels of pigment epithelium-derived factor (PEDF), a marker of insulin resistance, and the non-use of aspirin at baseline could “predict” the favorable response of the left main TBRmax (reduction from 2.17 to 1.93 (−12%), *p* = 0.014) to add-on therapy. Moreover, Δ120-min post-load plasma glucose and ΔPEDF were independent correlates of ΔLMT-TBR. Understandably, the authors concluded that 120-min post-load plasma glucose and PEDF values may be markers and potential therapeutic targets of coronary artery inflammation in T2DM [45].

In this section, there was a single animal FDG study. Virta et al. used a mouse model of atherosclerosis and T2DM to study the effects of the dipeptidyl peptidase-4 inhibitor linagliptin on atherosclerotic plaque and hepatic inflammation. PET imaging was not applied, but histology, radioactivity counting, and autoradiography of the excised aorta and liver demonstrated that 12 weeks of treatment with linagliptin improved glucose tolerance and reduced hepatic inflammation, but had no effect on the aortic plaque burden or atherosclerotic inflammation [44]. In the absence of a positive control, it is not possible to conclude whether linagliptin is not effective, or else whether the mouse is not a good model for human atherosclerosis.

In the remaining two human studies on FDG uptake, Ripa et al. conducted an RCT of 51 patients with T2DM of 10–12 years duration vs. 51 placebo-treated patients; they found that the insulin releasing drug liraglutide, administered at 1.8 mg/d for 26 weeks, had virtually no effect on FDG uptake in the carotids and ascending aorta, albeit an exploratory analysis suggested a borderline significant effect in individuals with a history of CVD [46]. The other human FDG study was a subanalysis of only eight male subjects with decreased insulin sensitivity from the Brucerius trial [25]. In these patients, Boswijk et al. observed that resveratrol, a polyphenol compound found in red grapes and blueberries and thought to be a powerful antioxidant, was associated with insignificantly higher FDG uptake in the carotids and aorta compared to placebo. In visceral adipose tissue, the increase in FDG uptake after resveratrol reached statistical significance (*p* = 0.024), and CRP-levels were not significantly affected by resveratrol treatment (*p* = 0.091) [49].

Zwakenberg et al. found in a randomized control trial with 35 T2DM subjects with CVD that six months of treatment with the vitamin K analog menaquinone-7 (360 µg/d) was associated with an insignificant change in femoral artery NaF uptake, compared to placebo. Calcification progression measured with CT was not reduced by vitamin K supplementation [43]. Finally, Bellinge et al. conducted a randomized control trial of 154 patients with self-reported, about 10-year lasting, unspecified diabetes and coronary calcification, as detected using CT, and found no effect of vitamin K1 or colchicine on coronary artery NaF uptake after three months, compared with the placebo [47]. However, a post hoc analysis of data, applying a modified dichotomous limit for increased NaF and individualized for each coronary artery, suggested that vitamin K1 supplementation was associated with decreased odds of developing new NaF PET-positive lesions in the coronary arteries [48].

## 3. Discussion

Only 28 articles were found suitable for analysis, despite the fact that early reports on FDG- and NaF-PET/CT imaging of atherosclerosis appeared more than 20 and 10 years ago, respectively [30,51]. The majority of studies focused on in vivo imaging of arterial wall inflammation with FDG, although it turns out that NaF is more consistently associated with cardiovascular risk factors and the probable precursor of CT-detectable calcification (Figure 3) [52,53,54]. All studies on NaF imaging except one [32] appeared during the last few years [35,38,39,40,43,47,48], suggesting that, until recently, the concept of inflammation as the initiator of atherosclerosis has been firmly entrenched in the minds of most diabetologists.

Correspondingly, a large proportion of studies cite inflammation as the cause of atherosclerosis, often with reference to old papers that did not actually prove this assumption, but rather present hypotheses sustained by a statistically indicated association between markers of inflammation and increased carotid intima-media thickness, PWV, CT-detectable arterial macrocalcification, or post-mortem-identified calcification [55,56,57,58,59,60,61,62,63,64,65]. None of the included 28 studies contained solid information about T1DM patients. Therefore, what is discussed below relates to pre-diabetic and/or T2DM subjects.

### 3.1. Disease Mechanisms and Targeting

Publications on this theme all used FDG and showed—despite differences in purpose and methodology—that arterial inflammation, assessed with FDG-PET, is a frequent occurrence, most pronounced in the lower parts of the aorta and present to a lesser extent upwards and downwards [27,34], albeit such that increased FDG uptake can be found in all major arteries including the coronary arteries. In this respect, the information about atherosclerosis in diabetes does not differ significantly from what is known from non-diabetics [4]. Clearly, inflammation is associated with the early stages of atherogenesis, and the uptake of PET tracers FDG and flouromethylcholine provide some measure of these early molecular signaling events. Despite the promise for the use of FDG, a serious limitation is the general nature of FDG as a glucose analog. Cells of the vascular wall in atherosclerosis, including vascular smooth muscle, endothelial, and immune cells, all have increased glucose uptake relatively proportional to the cellular metabolism. The most extreme example is vascular smooth muscle, which has been characterized as “glycolysis addicted” under healthy conditions, and increases glucose uptake substantially during normal contraction and during dedifferentiation and calcification in atherosclerosis [66]. Fluoromethylcholine might be more specific than FDG for inflammation, although the in vivo sensitivity seems modest [29]. It is tempting to look more closely at the circumstances that may play a special role for arterial FDG uptake in diabetics, including increased macrophage activity, altered cholesterol metabolism, and transport, and the fact that hyperglycemia and atherosclerosis share several mechanism at the molecular level [7]. Most of this is sparsely examined and poorly understood, and thus a more elaborate discussion at this stage can only be speculative.

### 3.2. Early Detection and Prevalence of Arterial FDG or NaF Uptake

With regard to prevalence, the present FDG studies suggest that arterial inflammation probably does occur more frequently in subjects with prediabetes and T2DM [31,33,34], whereas two studies with NaF showed somewhat contradictory results. Thus, the early study by Janssen et al. reported “linear” NaF uptake by visual inspection in the femoral arteries of 28/44 (64%) of elderly patients with unspecified diabetes who were examined for oncologic purposes [32], whereas the study Raggi et al. published six years later reported an increase in measured coronary artery NaF uptake in only 13 (15%) of 88 patients with long-lasting diabetes (48 T2DM and 40 T1DM), all of whom were asymptomatic for CVD [35]. The latter finding is in line with more recent studies challenging the notion that diabetes is an equivalent CVD risk, and that all diabetes patients are at high risk [67,68,69,70,71,72].

This reflects an ongoing lack of understanding regarding the link between ongoing inflammation, mirrored by increased FDG uptake, and active microcalcification, reflected by NaF accumulation. This is also illustrated also by the fact that multiple PET studies have demonstrated limited co-localization of arterial FDG and NaF uptake to be the norm rather than the exception [4]: see the example given in Figure 4. Possible interpretations of this missing link between FDG and NaF uptake are: (1) FDG uptake does not measure inflammation; (2) FDG does measure inflammation, but inflammation serves as a “trigger” of molecular signals for NaF binding and does not need to be present for continued NaF uptake; or (3) FDG does measure inflammation, but inflammation is not involved in microcalcification detected by NaF binding.

In contrast to the uncertainty of FDG measurements of inflammation, NaF uptake into tissue and the binding of cells is phenomenally solid. The dissociation constant (K_d_) for ^18^F-NaF binding to areas of calcification ex vivo is 0.6 pM, thus indicating that the ultra-high affinity [2] and specificity of NaF for binding hydroxyapatite is significantly greater than the binding of calcium bisphosphate, calcium pyrophosphate, or calcium oxylate [73].

### 3.3. Cardiovascular Risk in Diabetes

The two FDG studies available on this relationship were also partially contradictory. Strobl et al. examined four vascular beds (carotids, thoracic and abdominal aorta, and iliac arteries) in 315 patients scanned for non-CVD reasons, 15 of whom had unspecified diabetes, and found that increased blood-glucose-corrected FDG uptake was not statistically associated with diabetes in any of the four vascular beds, whereas CT-detectable calcification was associated with all four of them [36]. In contrast, Lee at al. observed in 290 subjects undergoing a general health check, of whom 92 had MetS, that blood-glucose-corrected carotid FDG uptake was significantly associated with cardiovascular risk factors, and that diabetes and MetS were independent determinants of high-carotid FDG uptake. Additionally, the Framingham Risk Score was similarly and significantly elevated in both subjects, without and with MetS, provided they had high instead of low carotid artery FDG uptake [37]. Finally, with regard to NaF, Takx et al. noted in subjects with T2DM and PAD that higher visual CT calcium in the femoral arteries, total cholesterol, and HbA1c were associated with higher femoral artery NaF TBRmax [38].

### 3.4. Disease Progression

The studies on progression could not explain a possible association between increased arterial FDG and increased NaF uptake. They suggested that the presence of microcalcification by NaF-PET is a precursor or predictor of upcoming cardiovascular events attributable to later-onset CT-detectable calcification, but whether this is actually the case is impossible to judge from the existing literature. However, it seems rather clear from PET studies of atherosclerosis in patients without diabetes [52,53,54], and now also from patients with prediabetes and T2DM, that increased NaF uptake in major arteries is associated with an increased tendency for later acute myocardial infarction and possibly peripheral vascular disease—major causes of death in diabetic patients [39,40].

### 3.5. Therapy

What remains to be discussed are aspects of therapy; these are important for our understanding because a positive therapeutic effect may bridge gaps of ignorance and sometimes point to unexpected disease mechanisms. This is what the three careful studies from the same Japanese institution did. Dealing with FDG uptake in the carotids, the ascending aorta, and the left main coronary artery, two of them demonstrated an FDG-reducing effect of about 10% following only four months of treatment with the insulin-sensitivity improving drug pioglitazone, whereas glimepiride, which stimulates pancreatic insulin production, had no effect [41,42]. As such, increased serum levels of pigment epithelium-derived factor (PEDF), a marker of insulin resistance and late post-load increases in plasma glucose during an oral glucose tolerance test, were identified as independently correlated to the reduction of NaF uptake in the left main coronary artery in T2DM patients [45].

Of other FDG studies of therapy, the Finnish study of atherosclerotic mice on a high-fat diet showed that added linagliptin improved glucose tolerance and reduced hepatic inflammation, but had no effect on the plaque burden or atherosclerotic inflammation, according to ex vivo examinations of the thoracic aorta and liver [44]. Among the human FDG studies, Ripa et al., in a randomized control trial of 102 patients with T2DM of a duration of 10–12 years, were not able to prove any changes in FDG uptake in the carotids and ascending aorta following a 26-week course of treatment with the insulin-releasing drug liraglutide. However, in an explorative analysis, they noted a potential effect (*p* = 0.052) in treated patients with a history of CVD [46]. In a sub-analysis of a former RCT, Boswijk et al. observed in eight male subjects with decreased insulin sensitivity that, after 34 days of placebo and resveratrol treatment (150 mg/day), carotid and aortic FDG uptakes were non-significantly higher, and visceral adipose tissue uptake was significantly higher after resveratrol compared to the placebo, while CRP-levels were not significantly affected [49].

Finally, among three NaF studies, Zwakenberg et al. found that, in T2DM patients with CVD, vitamin K supplementation for six months had no effect on femoral artery NaF uptake [43]. Similarly, Bellinge et al. studied 154 patients with coronary calcifications and long-lasting, self-reported, but unspecified T1DM or T2DM, that three months’ treatment with vitamin K1 or colchicine had no effect on NaF uptake in the proximal coronary arteries or the thoracic aorta [47]. Based on a later post hoc analysis of the same material, applying individualized but different cut-off limits for elevated NaF uptake in the coronary arteries, they stated that vitamin K1 supplementation independently decreased the odds of developing new NaF-PET-positive coronary and aortic lesions [48]. This study on therapy for coronary artery calcification clearly indicates that early intervention is necessary if it is to be effective. At the other extreme, there was an increase in coronary artery calcification with statin treatment, as assessed by intravascular ultrasound [74]. Early intervention points to the need for therapies that modulate the exosome release of components that contribute to the nidus of hydroxyapatite in the extracellular matrix [75], to which NaF binds.

Hence, it remains to be seen whether pioglitazone or another insulin sensitizer given for longer than four months can inhibit arterial inflammation in diabetics by more than the 10% shown in the Japanese studies; it is also unclear whether this type of drug can also reduce arterial wall NaF uptake and the later development of CT-detectable calcification. Elucidating these matters would significantly improve our understanding of atherosclerosis, and have a major impact on the treatment of atherosclerosis and its serious derivative diseases.

### 3.6. Limitations

Unfortunately, studies on PET imaging of atherosclerosis in DM are hampered by many of the same shortcomings as PET studies of atherosclerosis in general [4]. That is, there is no standardized methodology, and no consensus with regard to which measures are preferable for quantification or which arteries to focus upon. Readers looking for numbers characterizing the diagnostic accuracy of PET imaging of atherosclerosis will search in vain, because there is no unambiguous, generally accepted definition of atherosclerosis; however, in the minds of many, it is equal to the presence of increased lipid accumulation and/or macrocalcification in the arterial wall. Regarding the specificity of FDG imaging, there is no such thing, as increased FDG uptake mirrors increased glucose metabolism, a marker of multiple physiologic and pathophysiologic processes, the correct segmentation of which (in this case, in the arterial wall) determines the topographical ‘specificity’ of an abnormal finding.

When it comes to studies on diabetes, a particular variant was that 5 of the 20 studies reporting FDG uptake [25,33,37,45,47] used a correction for the pre-scan glucose concentration in diabetic subjects, as suggested by the EANM for oncologic scans [50] and later endorsed by an EANM position paper on PET imaging of atherosclerosis [76]. This correction made it impossible to compare the results of these four studies with the results of the FDG studies without correction. One of the five studies reported both corrected and uncorrected uptake values, and found small, but significant, differences in FDG uptake between T2DM and nondiabetic subjects; it also emphasized these data, neglecting the fact that there was no difference between the uncorrected values [25]. The three other studies reported corrected values only, leaving the reader unclear as to whether this correction was reasonable or not.

Many studies were retrospective, typically based on trials conducted for other purposes in patients with primary diseases other than diabetes, or were reanalyses of samples of other materials. There is low demand for prospective studies designed to study atherosclerosis in diabetes, and longitudinal studies with a follow-up period of more than a couple of years, which attempt to map the natural history of atherosclerosis in diabetes, are nonexistent. Hence, it is impossible to decipher the actual meaning or impact of the many reported significant “associations”, typically in the shape of a linear correlation, which was statistically significant at the 5% level, but often with a wide scatter, precluding solid assumptions about the underlying relationships. Some authors called associations of this kind ‘predictive’ of, for instance, future CV events, even though nothing can be stated for the individual subject with a reasonably high likelihood of, say, 85% or more. Therefore, the somewhat tedious conclusion is that a number of interesting observations have been made, but also that the interpretation of these findings is difficult and calls for new, more highly targeted, and better designed studies, the primary purpose of which is to illuminate the molecular mechanisms of diabetes.

### 3.7. Summary

With these reservations, the most striking findings in prediabetic and T2DM patients can be summarized as follows (see also Figure 5):Presumably, the same atherosclerosis mechanisms are at play in non-diabetic and diabetic patients, but NaF studies about this are lacking, as almost all the information comes from FDG studies.Arterial inflammation is very often present in diabetics, but the nature and frequency of the potential transition to microcalcification is not known, nor are the nature and frequency of the transition from microcalcification to CT-detectable calcification, although an increasing body of literature sustains the notion that NaF is a precursor for CT-detectable calcification.NaF uptake signals ongoing microcalcification, is associated with known cardiovascular risk factors, and appears to herald later cardiovascular events (e.g., myocardial infarction, stroke, PAD).Arterial inflammation in diabetics can be reduced by short-term treatment with the insulin sensitizer pioglitazone. However, it is unknown whether glitazone drugs can also reduce arterial NaF uptake, and this question deserves investigation in prospective longitudinal trials.

## 4. Materials and Methods

Using the principles of a systematic review including the Patient, Intervention, Comparison, Outcome Study (PICOS) approach [77], we searched PubMed/MEDLINE, Embase, and Cochrane Library with the strings given in the Appendix A. Extracted were peer-review articles in English published from 1 January 2000 to 31 March 2022, with no restrictions placed on comparator methods, outcome measures, or study design. Excluded were articles outside the scope of this review, editorials, letters, comments, conference proceedings, methodology studies, and all reviews and case stories. One experienced author (PFHC) reviewed the articles, extracting information on number, sex, age, type of patients, tracer (FDG/NaF), artery segment studied, purpose, quantification method, and the main findings.

## Figures and Tables

**Figure 1 ijms-23-10268-f001:**
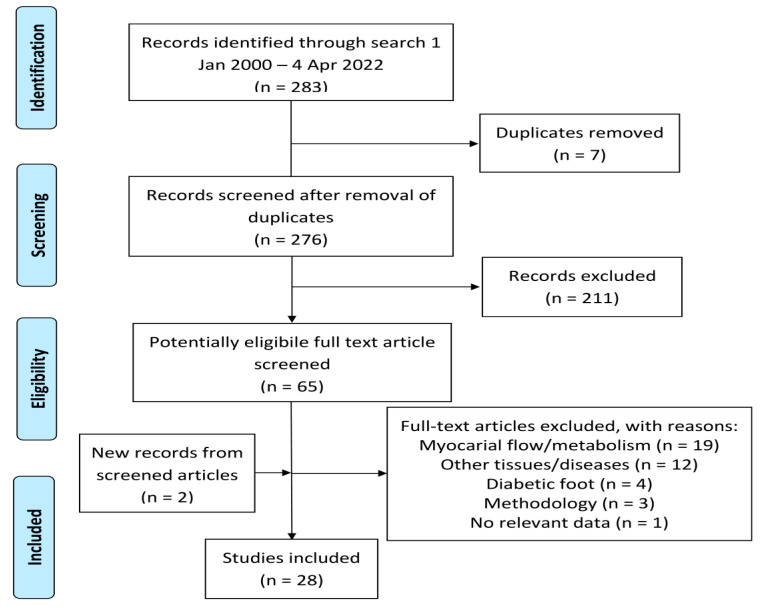
Modified PRISMA diagram showing literature selection.

**Figure 2 ijms-23-10268-f002:**
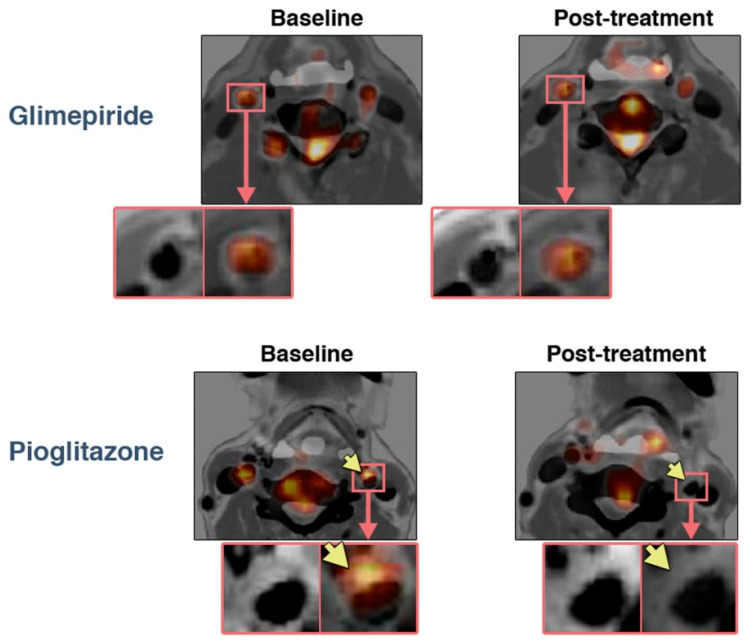
Treatment effects on FDG uptake in atherosclerotic plaques. Representative FDG PET/CT with contrast media images (**left**) at baseline and (**right**) after four months of treatment with (**bottom**) pioglitazone or (**top**) glimepiride. Note the reduction in FDG uptake in the atherosclerotic plaque with pioglitazone treatment (arrows). Reprinted with permission from Ref. [41]. © 2022, The American College of Cardiology Foundation.

**Figure 3 ijms-23-10268-f003:**
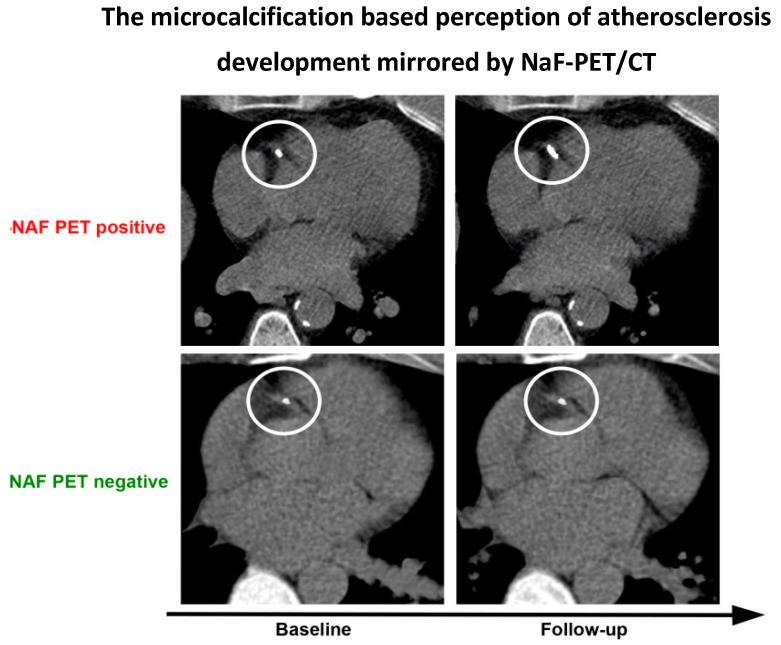
The recent ‘microcalcification-based’ perception of atherosclerosis development. Instead of showing inflammation, as FDG PET/CT is supposed to do, within the circles, this fabricated figure illustrates (**upper panel**) that a small, but NaF-avid (not shown directly) CT-detectable lesion in the proximal part of the right coronary artery has grown in size at follow-up after some years, at which point it has become more dense and less NaF avid due to a decrease in surface area. In contrast, (**lower panel**) a NaF-negative CT-visible lesion will usually not develop further. It remains unclear what the precise context for this phenomenon is; for details, see [52,53,54].

**Figure 4 ijms-23-10268-f004:**
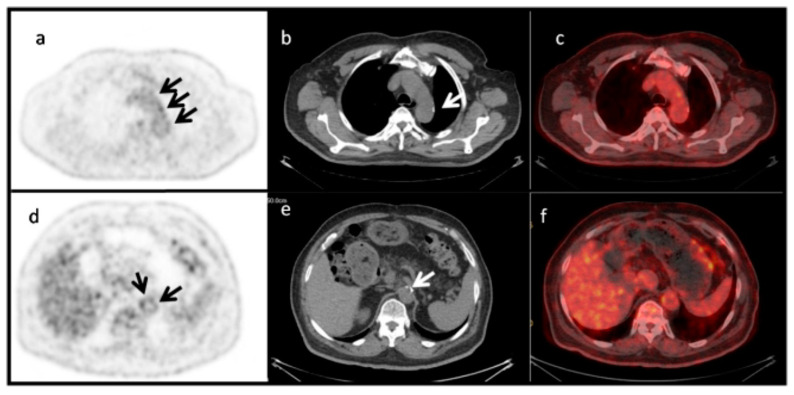
Limited co-localization of aortic wall FDG uptake and CT-detectable macrocalcification. PET images (**a**,**d**), CT images (**b**,**e**), and fused PET/CT images (**c**,**f**) of the aortic arch (upper panel) and abdominal aorta (lower panel) of a 72-year-old man with insulin-dependent diabetes. Macrocalcifications are visible in the low-dose CT images (white arrows in (**b**,**e**)), while faint and more pronounced NaF uptake in other locations is detectable by PET (black arrows in (**a**,**d**)). The SUVmax and SUVmean were 2.9 and 2.2 for the aortic arch, and 2.6 and 2.4 for the abdominal aorta, respectively. Reprinted with permission from Ref. [34]. © 2022, Hellenic Society of Nuclear Medicine.

**Figure 5 ijms-23-10268-f005:**
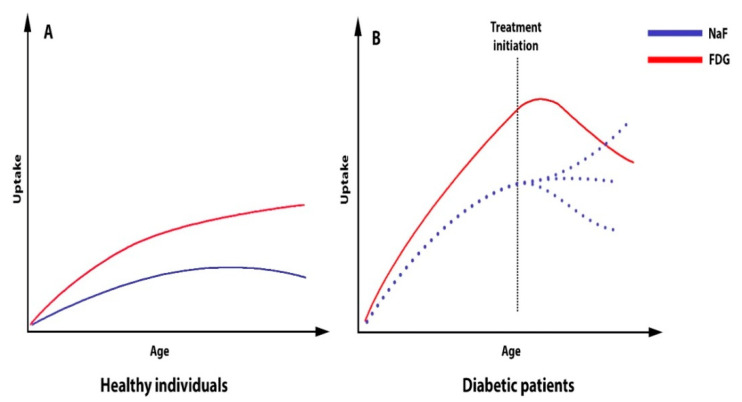
Schematic presentation of the arterial uptake of FDG and NaF in healthy individuals (**A**) and diabetic patients (**B**) as a function of age, as well as the effect of therapy (pioglitazone) on FDG uptake and the unknown effects (dotted lines) of any future attempts to influence NaF uptake.

## Data Availability

Not applicable.

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
