# Peer review of "Atherosclerosis Burdens in Diabetes Mellitus: Assessment by PET Imaging"

_ijms, 2022, doi:10.3390/ijms231810268_

Round 1

Reviewer 1 Report

This is an interesting review. The authors summarized both pre-clinical and clinical studies on the use of positron emission tomography and computed tomography (PET/CT) to detect early atherosclerosis change in diabetes mellitus, and showed the performances of FDG/NaF uptaking in diabetes-related atherosclerosis and after therapies. Overall, the manuscript was clearly written, and provided an updated information. I listed several concerns need to be addressed.

1. FDG uptake is a marker of metabolic activity and is used to measure the inflammatory state of several tissues. Would any other non-atherosclerosis tissue also take up FDG? How is the specificity of FDG uptake for the diagnosis of atherosclerosis?

2. For the readers’ benefit and interest, it is suggested to discuss more about the FDG uploading in macrophages (PMID: 33785060) in the part of inflammation mechanism and disease progression. It's because studies found that atherosclerosis is characterized by macrophages activity under hyperglycemia via NLRP3 inflammosome activity regulation (PMID: 30422379, 34197316) and cholesterol metabolism regulation (PMID: 32739420, 34436484, 29097366). This idea can help to explain how PET/CT reflects the pathophysiology of atherosclerosis on cellular and molecular level, and can also support the specificity of FDG uptake for atherosclerosis.

Author Response

Response to Reviewer 1 Comments

Point 1: FDG uptake is a marker of metabolic activity and is used to measure the inflammatory state of several tissues. Would any other non-atherosclerosis tissue also take up FDG? How is the specificity of FDG uptake for the diagnosis of atherosclerosis?

Response 1: The reviewer is right: FDG will mark increased glucose metabolism and therefore also ongoing inflammation in any tissue, meaning that FDG is an unspecific marker of inflammation due to several reasons, please see also Response 2. With regard to the specificity of FDG for the diagnosis of atherosclerosis, nobody can tell for several reasons included that there is no unambiguous definition of atherosclerosis. One may say that the “specificity” FDG has in the vascular wall is more a spatial one, meaning that the reliability and interpretation of an observed arterial FDG uptake depend on the accuracy with which non-vascular wall tissues are excluded by the used segmentation process, which varies very much from one lab to another. However, in principle, if vascular wall segmentation is 100% (which it will hardly ever be) increased arterial wall FDG uptake does mark increased vascular wall metabolism – whatever the reason for that is. We have shortly mentioned these uncertainties in eight additional lines in the limitation section on page 12 of the revised manuscript.

Point 2: For the readers’ benefit and interest, it is suggested to discuss more about the FDG uploading in macrophages (PMID: 33785060) in the part of inflammation mechanism and disease progression. It's because studies found that atherosclerosis is characterized by macrophages activity under hyperglycemia via NLRP3 inflammosome activity regulation (PMID: 30422379, 34197316) and cholesterol metabolism regulation (PMID: 32739420, 34436484, 29097366). This idea can help to explain how PET/CT reflects the pathophysiology of atherosclerosis on cellular and molecular level, and can also support the specificity of FDG uptake for atherosclerosis.

Response 2: We understand the Reviewer’s comment and do see hers/his point. However, once again it is not possible to elaborate more in-depth on this. Increased macrophage activity (as in adult-onset Still’s disease and e.g., in periodontitis of uncontrolled T2DM patients) may be an influential factor in vascular wall FDG uptake in atherosclerosis just as change in cholesterol and macrophage cholesterol metabolism together with change in cholesterol transport to and fro (PMID: 35163256) which may also be of some significance for the observed coming and going of arterial wall FDG uptake (PMID: 20562727) we have discussed in reference #4 of the present manuscript. However, unfortunately, all this is sparsely examined and poorly understood and thus a more elaborated discussion of this can only be speculative. The same goes for the potential role of hyperglycemia in atherosclerosis, since these two disease share several common mechanism at the molecular level (PMID: 31722564, 32155866), not to speak of the fact that the lumped constant is not constant for a specific tissue and may vary due to treatment (PMID: 31676728). These are more factors in play, and therefore, we have contented ourselves with inserting a few lines about this in section “3.1. Disease Mechanisms and Targeting” on page 10 in the revised version of the manuscript, hoping that the Reviewer can accept this “solution”.

Reviewer 2 Report

A systematic review focuses on the controversial and little-studied issues of
the role of imaging using positron emission tomography and computed tomography
of inflammatory atherogenic changes in the vascular wall in diabetes mellitus. The review is well written. I have no comments.

Author Response

Response to Reviewer 2 Comments

Point 1: A systematic review focuses on the controversial and little-studied issues of the role of imaging using positron emission tomography and computed tomography of inflammatory atherogenic changes in the vascular wall in diabetes mellitus. The review is well written. I have no comments.

Response 1: We thank this Reviewer for her/his kind words and have no further comments.

We thank this reviewer for hers/his kind words and have no further comments.